# Training in Aortic Arch Surgery as a Blueprint for a Structured Educational Team Approach: A Review

**DOI:** 10.3390/medicina59081391

**Published:** 2023-07-29

**Authors:** Philipp Discher, Maximilian Kreibich, Tim Berger, Stoyan Kondov, Matthias Eschenhagen, David Schibilsky, Magdalena Bork, Tim Walter, Salome Chikvatia, Roman Gottardi, Bartosz Rylski, Matthias Siepe, Martin Czerny

**Affiliations:** 1Department of Cardiovascular Surgery, University Heart Centre, University Hospital Freiburg, 79106 Freiburg, Germany; philipp.discher@uniklinik-freiburg.de (P.D.);; 2Faculty of Medicine, Albert-Ludwigs-University of Freiburg, 79110 Freiburg, Germany; 3Department of Cardiac Surgery, University Heart Centre, University Hospital Bern, 3010 Bern, Switzerland

**Keywords:** training, teaching, aortic arch surgery, frozen elephant trunk (FET), endovascular

## Abstract

*Background and objectives:* The treatment of pathologies of the aortic arch is a complex field of cardiovascular surgery that has witnessed enormous progress recently. Such treatment is mainly performed in high-volume centres, and surgeons gain great experience in mastering potential difficulties even under emergency circumstances, thereby ensuring the effective therapy of more complex pathologies with lower complication rates. As the numbers of patients rise, so does the need for well-trained surgeons in aortic arch surgery. But how is it possible to learn surgical procedures in a responsible way that, in addition to surgical techniques, also places particular demands on the overall surgical management such as perfusion strategy and neuro-protection? This is why a good training programme teaching young surgeons without increasing the risk for patients is indispensable. Our intention was to highlight the most challenging aspects of aortic arch surgery teaching and how young surgeons can master them. *Materials and Methods:* We analysed the literature to find out which methods are most suitable for such teaching goals and what result they reveal when serving as teaching procedures. *Results:* Several studies were found comparing the surgical outcome of young trainees with that of specialists. It was found that the results were comparable whether the procedure was performed by a specialist or by a trainee assisted by the specialist. *Conclusions:* We thus came to the conclusion that even for such a complex type of intervention, the responsible training of young surgeons by experienced specialists is possible. However, it requires a clear strategy and team approach to ensure a safe outcome for the patient.

## 1. Introduction

The training of new cardiovascular surgeons is an increasing challenge to the medical system and the physicians working in it. As in any other discipline, it is important that young doctors can learn from the experience, knowledge and acquired skills of older specialists. However, surgical training requires very intensive practical training in order to acquire the necessary skills. This is becoming more and more challenging due to the following aspects.

Interventions in the thoracic aorta are developing rapidly. Today, new surgical methods and devices allow the treatment of significantly more complex pathologies and thus a significantly larger number of patients [1,2]. At the same time, the number of patients is also rising due to increasing life expectancy, closer monitoring by screening programs and follow-up examinations by specialists. This has resulted in an increasing demand for well-trained surgeons who are able to perform such procedures safely. On the other hand, there has been a reduction in the number of hours available for training due to new occupational health and safety laws.

In order to meet these requirements and to enable the responsible and effective training of young surgeons, a well-thought-out training program is needed.

The aim of this review is to contribute to the creation of such a program by searching the literature for articles dealing with training in aortic arch surgery in particular. This should help in the first instance to gain an understanding of which demands a young surgeon needs to meet. Furthermore, this paper serves as an overview of the extent to which young surgeons without much experience in complex aortic surgeries can perform them as operators without endangering the patient. Finally, specific measures that help to improve training in aortic arch surgery will be presented.

## 2. Increasing Numbers and Technical Development of Aortic Arch Surgery

Surgery of the aortic arch (AA) is one of the newer fields in cardiovascular surgery, and it offers a huge variety of technical options [1,2]. Due to its rapid development, the number of AA procedures has risen rapidly during the last decades. Nevertheless, it remains a highly demanding procedure carrying potential risks. To ensure that a sufficient number of surgeons can perform these operations and reduce the ultimate risk to a minimum, the structured, high-quality education of young trainees is fundamental.

Procedures have developed technically from classical surgery mainly using the island technique and performing the descending anastomosis in zone 3 to the selective reimplantation technique of the supra-aortic branches with the descending anastomosis in zone 2 where the frozen elephant trunk (FET) technique has become the standard approach in the vast majority of patients, thereby already providing a platform for potential open or endovascular downstream aortic repair [2,3,4]. Likewise, transcatheter techniques for AA repair, mainly branched endovascular AA repair, have gained momentum [5]. These operations are here to stay, but their broader application remains restricted by anatomical and material limitations. Supra-aortic transpositions are still being performed but are usually limited to carotid-to-subclavian artery bypass or transposition and, in certain patients, double transposition [6,7]. 

Cannulation techniques for cardiopulmonary bypass (CPB) have also changed, where subclavian cannulation is now the approach of first choice. There are alternative cannulation sites depending on the operation and underlying pathology [8,9].

## 3. Demands on the Surgeon in Aortic Arch Surgery

The challenge of aortic surgery starts already long before the operation begins. Decision-making about the extent of aortic replacement and the choice of open, endovascular or hybrid treatment requires a profound understanding of the underlying disease. The intervention’s extent must be determined with anticipatory consideration, taking into account the accompanying circumstances and the surgeon’s personal surgical experience and skills. In concrete terms, this usually means having to decide between a partial arch replacement with an open anastomosis on the one hand and a total arch replacement via the FET approach on the other [1,10]. Thoracic endovascular aortic repair (TEVAR) combined with carotid-to-subclavian artery bypass is an appropriate option for certain patients [6]. For those requiring full arch replacement but who are unsuitable for open surgery, also total endovascular arch repair (TER) should be considered [11]. Finding the best strategy represents a major challenge, especially in the case of emergency interventions. 

In patients undergoing open surgery, the perfusion strategy plays a key role in their outcome. With its roughly 5% frequency, stroke is one of the most severe complications after AA surgery [12,13]. To minimise the remaining risk, a clear and structured algorithm for cerebral perfusion is essential. Unilateral cerebral perfusion via the right subclavian artery and right carotid artery by clamping the brachiocephalic trunk during lower body hypothermic circulatory arrest (HCA) can be seen as the least common denominator in a standard perfusion protocol. In cases of insufficient collateralisation through the circle of Willis, or a longer expected selective antegrade cerebral perfusion (SACP) time, bilateral cerebral perfusion by additionally cannulating the left carotid artery is known to enable an excellent outcome [14]. In rare cases of the vertebral artery’s unilateral dominance for cerebellar perfusion (PICA constellation), even a third perfusion line, addressing the left subclavian artery (LSA) should be considered to prevent neurological complications affecting the cerebellum (trilateral antegrade cerebral perfusion). Cerebral perfusion protocols may also need to be modified in patients presenting an isolated origin of the vertebral artery from the AA. Thus, a wide range of clinical scenarios must be anticipated, and surgeons need to adapt to the individual situation [15].

The main technical requirement is to manage several anastomoses in a predefined order, where LSA access can be particularly challenging. The patient’s underlying pathology and associated tissue quality are also decisive. Whenever possible, action should be taken to facilitate surgery. A noteworthy easement is to proximalise the descending anastomosis into Ishimaru zone 2 by transection and the oversewing of the native origin of the LSA before completing the anastomosis [1]. This simple trick enables better surgical access and exposure of the descending anastomotic site and lowers the risk of laryngeal nerve injury. Shorter lower-body HCA times are also more likely [16,17]. 

In addition to such classical surgical capabilities, endovascular skills are essential when it comes to the FET procedure [18]. Maneuvering the stent-graft portion into the right place in the descending aorta requires the appropriate handling of the stent-graft portion of the prosthesis and if needed, employing additional tools such as a transfemoral wire insertion to navigate into the true lumen, as well as intraoperative angioscopy [19].

Another key factor in AA surgery is the procedure time. The durations of lower body HCA, SACP time and CPB time must all be kept to a reasonable minimum to avoid collateral injury.

In summary, all these factors make the surgical treatment of AA a demanding procedure in many respects–most of which require structured stepwise training.

## 4. Patient Outcome and Results in Teaching vs Non-Teaching Operations

In the context of improving training concepts for complex AA surgery, clinicians need clear feedback on the safety and accountability of teaching procedures in the clinical context. Although there is much discussion about efficient and safe teaching in AA surgery, there is a paucity of reliable data actually assessing teaching procedures. The main focus of such investigations must clearly be put on patient outcomes. There are additional factors deserving consideration, such as economically key parameters like procedural times and patients’ length of hospital stay.

Recent studies have focused on these key points and have led to conclusions with a potential impact on future teaching concepts for AA surgery.

A recent study analysed the impact of the surgical team’s composition on patient outcomes in AA replacement using the FET technique [20], Figure 1. Each surgeon was rated according to his individual experience with the FET procedure, and a team score was devised by adding up the scores of the operator and the assistant. The authors then determined whether the total team score or the team’s individual composition (either two surgeons with an equivalent score or one more experienced with one less experienced) had an influence on the outcome. To do that, the authors compared mortality rates, takeback for bleeding and neurological complications such as stroke and spinal cord injury among different team compositions. 

They found that a team below a certain score had a higher risk of complications than teams above that threshold score. Upon closer inspection, they discovered that whether the more experienced team member performed the operation or assisted the less experienced colleague had no effect on the surgical outcome.

These observations led the authors to conclude that it is responsibly justifiable to let a less experienced trainee carry out AA replacement as long as an experienced specialist assists during the procedure. 

This is definitely a fundamental point, as patient safety is doubtless the ultimate goal of any intervention, and concern about it can quickly lead to disregarding the idea of training young assistant surgeons. There are certainly other important factors to consider when deciding about teaching procedures. But awareness of how important team composition is should definitely be incorporated within such considerations, thereby encouraging the training of upcoming surgeons even in such complex interventions.

Another study compared endovascular aortic repair procedures (EVAR) for abdominal aortic aneurysms performed either by an endovascular specialist or by a trainee under supervision by a specialist [21], Figure 2.

They observed no relevant differences regarding EVAR procedure outcomes whether performed by specialists or trainees in terms of complications such as early mortality, endoleak or reintervention rate. Furthermore, the doses and X-ray times needed for the procedure did not differ between the two groups. What differed, however, were the total intervention time and the patient´s postoperative in-hospital stay: both were longer in patients treated by a trainee.

These observations concur with others conducting a similar study involving open heart surgery [22]. They detected longer operation times and longer extracorporeal circulation times for the trainee group, whereas the incidence of bleeding, neurological complications, reoperation or in-hospital mortality did not differ significantly in patients operated on by a specialist or a trainee.

All these findings are consistent irrespective of the procedure regarding patient safety [20,21,22]. But at the same time, extended intervention times are another challenge that is often decisive when selecting an operator in the clinic. Both the economic aspect of a very tightly scheduled operation planning and the increased burden on the whole team must be compensated for by making certain adjustments. 

Furthermore, a major study showed better AA surgery outcomes in teaching hospitals than in non-teaching hospitals [23]. 

A teaching hospital is more likely to have established clear guidelines for treating certain cases, whereas in non-teaching hospitals the exact treatment regime may be more influenced by the operator’s personal preference. This would mean that similar procedures are done more frequently, and thus a higher level of surgical skills should be achievable and maintained in teaching hospitals. Especially for trainees undergoing education in AA surgery, it makes sense to train in one uniform procedure, clarifying each necessary step. This approach would help prevent confusion through too many confounding factors caused by the responsible specialist’s overly individualized approach. A uniform setting is more beneficial for both the trainer and the trainee.

There are no comparable studies examining the teaching of interventions in TER. However, on the basis of the aforementioned findings, it seems conceivable that a trainee being supervised by a specialist can also perform such interventions safely.

## 5. Suggestions for Teaching Concepts in Aortic Centres

The conscientious education of surgical trainees is definitely possible but requires a dedicated, structured approach from the trainee and teaching specialist. It is essential to make the setting for teaching as agreeable as possible for the supervising specialist. A key step in that direction is a clear and reproducible teaching concept. It should not be up to the supervising specialist to think of a teaching strategy before each procedure but rather to follow a set sequence of educational steps. These steps should be defined by a team, optimally consisting of the department head, the senior surgeon in charge of the teaching programme and a representative of the trainees. Together they should develop a training plan that specifies the different steps to be learned to acquire surgical skills. Key to developing solid surgical skills is the performance review and constructive evaluation of one´s results. Keeping a logbook is a very helpful tool for recording all this valuable information.

Essential in such a curriculum is a fixed rotation schedule for the aortic team. This ensures that the trainee is present for the maximum number of aortic procedures. Only through regular exposure to and confrontation with such complex diseases and interventions is solid skill acquisition possible.

Furthermore, having the aortic team hold regular internal meetings to discuss all potential aortic interventions would be very beneficial. Together with an aortic outpatient clinic attended regularly by the trainee, this would contribute to a fundamental and thorough understanding of the different pathologies and their specific treatment options. This is essential to make the right choice among the many open surgical and interventional treatment options available.

Another helpful tool would be assigning a permanent mentor to each trainee, which would benefit both: the trainee gets to know all the detailed procedures in a set way and can thus gain confidence more quickly. The teaching specialist, on the other hand, can better assess the protégés’ abilities and learn how much he can expect out of him and in which situations extra support is needed. 

When it comes to procedural skills, it helps to standardize the surgical process to ensure that key steps are taken in the same manner regardless of which departmental specialist taught them. This allows the trainee to gain experience faster with the variety of demanding steps and focus more on the technical challenges. To make surgery more pleasant and feasible for both the trainee and the teaching specialist, the procedures must be learned gradually. Learning the technical requirements stepwise, starting with cannulation and establishing CPB and then technically clean prosthesis implantation and flawless hemostasis–all these surgical steps help to ensure the greatest possible safety and efficacy. Each step can be checked by means of a logbook, and individual feedback can accelerate the learning process.

To improve a key aspect of AA procedures, namely neuroprotection, a standardised protocol set preoperatively would help surgeons choose the ideal perfusion regime. By relying on CT angiography and intracerebral vascular ultrasound, circulatory deficits during extracorporeal circulation can be anticipated and prevented by well-adjusted cannulation. 

Another topic that is definitely worth looking at a bit closer is simulated training in learning surgical techniques. This particular form of training is very common in other fields of high performance such as aviation, military or professional sports to train the necessary skills in a predefined setup before performing under real and more uncertain conditions. This method brings along a number of advantages.

In order to point out the impact of simulated training on the performance of a surgical trainee, members of the European vascular workshop have been assessed after undergoing a simulation program for abdominal aortic aneurysm repair regarding different aspects [24]. Those consisted of procedural time, technical skills and procedural skills. This study was able to show that the procedural result was significantly better and the procedural time significantly lower after the simulation program compared to the results before the program. One detail that is worth pointing out is the fact that trainees with lower performance scores before undergoing a simulated training showed the biggest capacity of improvement undergoing the training program. This indicates that simulated training can show a great benefit to the surgeon’s abilities, particularly in an early stage of surgical education. 

The biggest and most obvious advantage of simulated training is the patient’s safety. The risk for serious complications in an operation performed by a trainee can be reduced if routine and safety for the procedure have been gained in a simulation model before operating on a patient.

Furthermore, it is beneficial that the procedure can be split up into different steps. This makes it easier to learn the surgical procedure’s different steps more precisely and in better quality. It also allows the trainee to work in a more targeted way on special deficits and train a much higher volume of the steps which require improvement without spending too much time on the whole procedure.

Particularly difficult situations or even complications can also be simulated in such a model. This can be a great advantage because such situations rarely occur in everyday clinical practice, but in an emergency, it can make a decisive difference whether this situation has already been practiced or whether the surgeon is facing this challenge for the first time.

Even though a surgeon must be able to perform at his best under conditions of high pressure and stress, the learning curve can be much steeper when training under circumstances with less stress and the possibility of detailed teaching. 

A well-established simulation setup can additionally serve to train new techniques for more experienced surgeons or to try out new devices. Furthermore, teamwork, as well as leadership qualities for more experienced surgeons, can be trained in those models. In this way, it can not only support the training of young surgeons but also contribute to the development of cardiothoracic surgery in general.

Surgery has always been the most practical discipline in medicine. Since practical skills can only be acquired through hands-on practice, a high number of repetitions is a crucial factor in the training of a surgeon. Procedures that could otherwise only be performed by specialists practicing for years can thus be learned and applied in earlier years.

In conclusion, it can be said that training under simulated conditions would therefore be particularly useful for cardiovascular surgery, in which an ever-increasing number of different and highly complex procedures have to be performed in a reliable manner. In other disciplines, such as emergency medicine, this is already a standard part of training.

There are various approaches for realizing such a simulation program. On the one hand, operations can be performed on cadavers; on the other hand, there are also artificial models of high quality on which an operation can be simulated. Since particularly expensive devices can be used for training several times in such a setup, this also offers an economic advantage. Support by the industry seems conceivable since medical technology companies have an interest in making their own products familiar to upcoming surgeons.

The last key aspects that should not be forgotten when talking about developing all the necessary skills for being a great aortic surgeon are his mindset and mental skills.

These soft skills are of central importance to a successful operation due to several reasons. Since the psychological pressure caused by responsibility and one’s own ambitions as well as the prevailing competition can be stressful, resilience against these influencing factors is essential. Frustration tolerance is also necessary during lengthy training until the required surgical skill is achieved. 

It is, however, difficult to assess these skills objectively and to train them specifically. Personal predisposition undoubtedly plays a major role. Such factors can also be trained, at least to a certain extent, provided they are recognized in a targeted manner.

The authors of a recent review concluded that different training methods for mental skills can have a positive impact on surgical performance under both simulation and real conditions [25].

The number of training options is almost as varied as the number of skills to be trained. The best-known methods in this regard include relaxation techniques, visualization and conscious self-assessment. 

The need, openness and interest in such training vary greatly between different surgeons. In addition to the complexity and difficulty of grasping this topic, prejudices certainly also play a role. It is often misunderstood as a weakness when one seeks help from professional mental trainers, although this has long been common practice in professional sports or even for special forces in order to be able to perform better under pressure and high stress. Even if this approach has so far been a rarity in surgical training, it would certainly be a useful addition to the existing training in the future.

In summary, the evidence to date clearly indicates that effective training of surgeons for AA procedures is possible without increasing the risk for the patients. However, to guarantee this in everyday clinical practice, certain precautions must be taken to enable the training process to take place on a scale ensuring a sufficient supply of aortic specialists for the growing numbers of future cases.

In addition to the practical tools mentioned above, structural adjustments are certainly necessary. Since the training of young surgeons is a fundamental task of university hospitals in addition to patient care, this must also be considered in the provision of resources and personnel. Politicians and hospital management must ensure that training does not fail because of departmental overload or economic factors. To support training, individual remuneration for teaching interventions could also be considered, for example, as these kinds of interventions place greater demands on the department.

At the end of the day, education remains a challenge on many different levels, and it can only be mastered if we work together. However, the present results should definitely encourage the intensification of educational efforts among surgical teams.

## Figures and Tables

**Figure 1 medicina-59-01391-f001:**
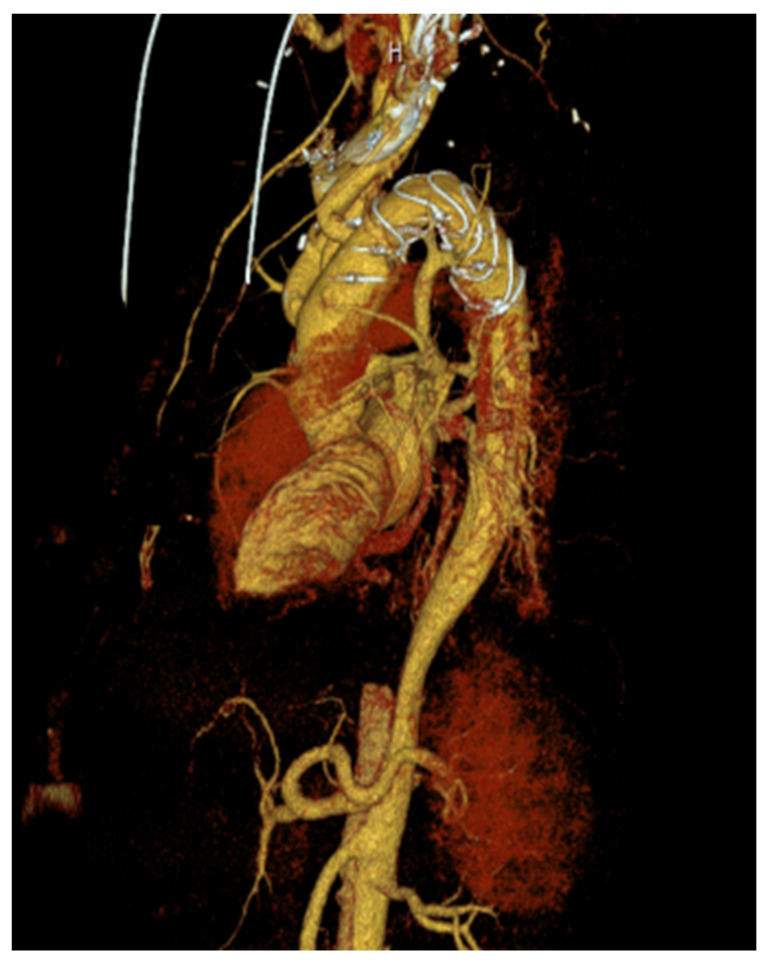
Representative example of a frozen elephant trunk (FET) procedure as a teaching case: 53-year-old male patient presenting with an acute Type Non-A-Non-B aortic dissection. Mild coronary sclerosis, no other pre-existing diseases.

**Figure 2 medicina-59-01391-f002:**
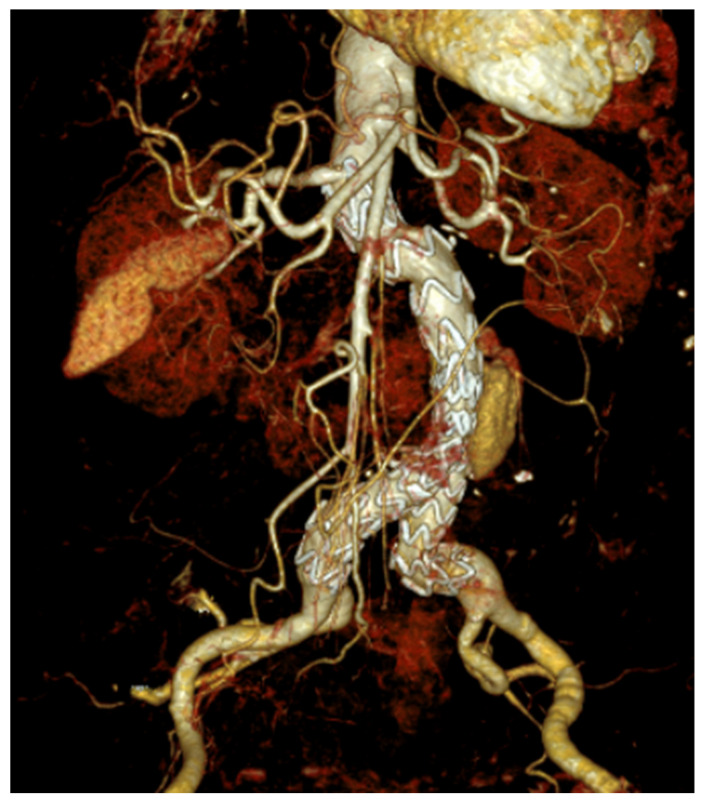
Representative example of an endovascular aortic repair (EVAR) procedure as a teaching case: 78-year-old female patient presenting with an infrarenal aortic aneurysm. Coronary two-vessel disease. No other pre-existing diseases.

## Data Availability

No new data was collected in the course of this work, only existing studies were cited. These are referred to in the references section.

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
