# Peer review of "Training in Aortic Arch Surgery as a Blueprint for a Structured Educational Team Approach: A Review"

_medicina, 2023, doi:10.3390/medicina59081391_

Round 1

Reviewer 1 Report

The article aims to demonstrate how a good training program is important in teaching young surgeons aortic arch surgery without increasing risks for patients.

While the first paragraph, a general overview of main techniques of aortic arch surgery are mentioned without going into depth; the second paragraph points out the difficulties of aortic arch surgery, including the choice of open, endovascular, or hybrid treatment, and the decision-making process for the perfusion strategy technique, among other factors.

Several interesting studies are cited in the third paragraph, although briefly, demonstrating that the risk to patients is not increased when a less experienced surgeon is assisted by an experienced one.

In the first study, mortality rates, instances of bleeding, and neurological complications were evaluated, which are certainly important factors but not the only ones to be considered. The duration of the surgery and postoperative hospitalization also play a significant role, both for the patient's well-being and from an economic standpoint.

The article referenced in line 190 is indeed interesting as it highlights the significance of established guidelines in improving outcomes in aortic arch surgery. This topic is further expanded upon in the subsequent paragraph, where the authors propose various suggestions to enhance the teaching approach in aortic centers, with a focus on the importance of a structured procedure.

Regarding the concept mentioned in line 231, it is certainly valuable to assign a permanent mentor to each trainee. This allows the trainee to gain confidence quickly, and enables the mentor to develop a deeper understanding of their student, determining the appropriate moments to provide assistance. However, it is crucial for the trainee to also learn alternative approaches when faced with challenges. While following a predetermined lineup during the surgical procedure facilitates gradual learning and focuses on specific technical challenges, it is equally important to be open to different possibilities and approaches when encountering difficulties.

One issue that is rarely mentioned in scientific literature is the psychological aspect of carrying-out AA operations and how to train it. In detail, it is reasonable that a model student after a complete period of surgical training, could, standing in front of his teacher, complete almost all cardiac operations without weigh-on results. A different topic is when and how it should, and if really everybody could, became a first surgeon having in front of any kind of second surgeon and in all operative settings. In the latter, the most important tool is how to train and build the personality of the trainee; therefore, having the same good results in the future.

Overall, the article has offered an interesting perspective on the discussed topic. The presented arguments are compelling and supported by solid evidence. In general, the article has generated curiosity and has encouraged further exploration.

Only minor revision requested.

Author Response

Dear reviewer,

thank you for your comments.

Comment: One issue that is rarely mentioned in scientific literature is the psychological aspect of carrying-out AA operations and how to train it. In detail, it is reasonable that a model student after a complete period of surgical training, could, standing in front of his teacher, complete almost all cardiac operations without weigh-on results. A different topic is when and how it should, and if really everybody could, became a first surgeon having in front of any kind of second surgeon and in all operative settings. In the latter, the most important tool is how to train and build the personality of the trainee; therefore, having the same good results in the future.

Answer: This is definitely a very interesting and crucial aspect in the development of a surgeon. We have therefore added another paragraph to our article on this subject.

Changes made: We inserted the available information (page 10, lines 337 )

Reviewer 2 Report

Dear Authors, 

You should clarify the concept of training programs in aortic arch surgery. It is not clear if this is a message to surgeons, perfusionists, anesthesiologists or cardiologists? The format of the article does not allow a global look at this problem. Therefore, the authors should decide on the target audience. And then form the concept of training in the aortic arch surgery.

Author Response

Dear reviewer,

thank you for your comments.

Comment: You should clarify the concept of training programs in aortic arch surgery. It is not clear if this is a message to surgeons, perfusionists, anesthesiologists or cardiologists? The format of the article does not allow a global look at this problem. Therefore, the authors should decide on the target audience. And then form the concept of training in the aortic arch surgery.

Answer: The article is primarily aimed at surgeons. It is about presenting the challenges for a young surgeon, especially in aortic arch surgery, and deriving from this how the necessary skills can be learned. on the basis of the available literature, suggestions are made as to how training in this area can be optimized despite growing demands. An exact training concept should be defined individually by each clinic.

Changes made: No changes made